# A Self-Assembling Pfs230D1-Ferritin Nanoparticle Vaccine Has Potent and Durable Malaria Transmission-Reducing Activity

**DOI:** 10.3390/vaccines12050546

**Published:** 2024-05-16

**Authors:** Nichole D. Salinas, Rui Ma, Holly McAleese, Tarik Ouahes, Carole A. Long, Kazutoyo Miura, Lynn E. Lambert, Niraj H. Tolia

**Affiliations:** 1Host-Pathogen Interactions and Structural Vaccinology Section, Laboratory of Malaria Immunology and Vaccinology, National Institute of Allergy and Infectious Diseases, National Institutes of Health, Bethesda, MD 20892, USA; nichole.salinas@nih.gov (N.D.S.);; 2Vaccine Development Unit, Laboratory of Malaria Immunology and Vaccinology, National Institute of Allergy and Infectious Diseases, National Institutes of Health, Bethesda, MD 20892, USA; 3Laboratory of Malaria and Vector Research, National Institute of Allergy and Infectious Diseases, National Institutes of Health, Rockville, MD 20852, USA

**Keywords:** single-copy nanoparticle vaccine, Pfs230D1, malaria, ferritin

## Abstract

Malaria is caused by eukaryotic protozoan parasites of the genus *Plasmodium*. There are 249 million new cases and 608,000 deaths annually, and new interventions are desperately needed. Malaria vaccines can be divided into three categories: liver stage, blood stage, or transmission-blocking vaccines. Transmission-blocking vaccines prevent the transmission of disease by the mosquito vector from one human to another. Pfs230 is one of the leading transmission-blocking vaccine antigens for malaria. Here, we describe the development of a 24-copy self-assembling nanoparticle vaccine comprising domain 1 of Pfs230 genetically fused to *H. pylori* ferritin. The single-component Pfs230D1-ferritin construct forms a stable and homogenous 24-copy nanoparticle with good production yields. The nanoparticle is highly immunogenic, as two low-dose vaccinations of New Zealand White rabbits elicited a potent and durable antibody response with high transmission-reducing activity when formulated in two distinct adjuvants suitable for translation to human use. This single-component 24-copy Pfs230D1-ferritin nanoparticle vaccine has the potential to improve production pipelines and the cost of manufacturing a potent and durable transmission-blocking vaccine for malaria control.

## 1. Introduction

Vaccines have proven to be highly effective for viral and bacterial diseases; however, the development of a vaccine for malaria has been complicated. Malaria, caused by the eukaryotic protozoan parasite of the *Plasmodium* genus, still lacks a highly effective vaccine and there are 249 million new cases a year and 608,000 deaths annually [1]. Tragically, the majority of deaths are in children under the age of 5 [1,2,3]. Recently, two vaccines against malaria, RTS,S/AS01 and R21/Matrix-M, have been approved by the World Health Organization, and rollout of the vaccines has begun in many endemic countries [1]. Unfortunately, both vaccines have been approved for use only in young children, and the large number of vaccinations necessary for both vaccines poses a challenge for vaccine compliance while leaving a large portion of the population susceptible to malaria. This underscores the necessity of developing new malaria vaccines.

Malaria vaccines can be broken down into liver stage (LSVs), blood stage (BSVs), or transmission-blocking vaccines (TBVs). LSVs include RTS,S and R21 [2,3] and target sporozoites to prevent invasion of the liver or development of the parasite within liver cells. LSVs can also prevent parasite transition in the human host from the liver stage to the erythrocytic cycle and block the development of clinical symptoms. BSVs target the erythrocytic cycle of the parasite wherein parasites undergo multiple rounds of asexual replication within red blood cells (RBCs), ultimately resulting in lysis of the infected RBCs. The erythrocytic cycle results in the clinical symptoms of malaria, and BSVs either reduce or block disease at this stage. TBVs target the sexual stage of the parasite that occurs within the mosquito vector [4,5]. A small percentage of parasites convert to male and female gametocytes during the erythrocytic cycle. These gametocytes are then taken up by a mosquito during a blood meal and further undergo sexual replication within the mosquito midgut to generate sporozoite-stage parasites that can infect humans. While LSVs and BSVs benefit the individuals who receive them, TBVs prevent the transmission of the disease mediated by the mosquito vector from one human to another [4,5].

The leading TBV candidates in clinical trials include Pfs230 [6,7,8,9,10,11,12,13,14,15,16,17,18,19] (NCT02942277, NCT05135273, and NCT02334462), Pfs25 [15,16,20] (NCT02942277, NCT00295581, NCT01867463, NCT02013687, NCT02532049, NCT02334462, NCT04130282, and NCT04271306), and Pfs48/45 [8,10,12,21,22,23,24] (NCT05400746). Pfs230 [6,7,8,9,11] is a gametocyte surface protein that is part of the 6-cysteine protein family that includes the sexual stage proteins Pfs48/45, Pfs47, Pfs230p, Pfs36, and Pfs38 and the asexual stage proteins Pf41 and Pf12 [9,12,25]. All these proteins contain domains characterized by conserved cysteines [6,7,8,9,11,26]. Pfs230 is a 360 kDa protein that is processed during gametocytogenesis into a ~310 kDa form that contains an unstructured N-terminal region followed by 14 6-cysteine domains arranged in 7 pairs of A-type and B-type 6-cysteine domains (Figure 1A) [6,7,8,9,11,12,26]. Pfs230 plays a role in the exflagellation of male gametes in the mosquito midgut, and disruption of Pfs230 results in a reduction in oocysts following fertilization [10,12,27]. Vaccination with Pfs230 induces antibodies with transmission-reducing activity (TRA), as measured by the standard membrane feeding assay (SMFA) [11,14,26,28,29,30,31,32]. High TRA is dependent on vaccination with a segment of Pfs230 comprising the unstructured N-terminal region and domain 1 (D1) of Pfs230 and requires human complement for function [11,14,26,28,29,30,31,32]. The human antibody response to vaccination with Pfs230D1 also established that one face of Pfs230D1 elicits potent transmission-blocking antibodies, while the opposing face elicits limited or no transmission-blocking activity [33].

Pfs230D1 is a small protein and suffers from poor immunogenicity in certain adjuvants due to its size. To increase size and immunogenicity, Pfs230D1 has been conjugated to carrier proteins such as ExoProtein A (EPA) [14,16,34], liposomes [35,36], and outer membrane vesicles [37] and incorporated into virus-like particles (VLPs) [28,38] to form two-component nanoparticles. All of these systems require the generation of multiple components that need to be assembled prior to immunization, complicating vaccine production and increasing cost [39]. Recent advances have been made in the creation of single-component nanoparticles consisting of a genetic fusion of an antigen to bacterial proteins that naturally form nanoparticles such as ferritin, lumazine synthase, and dihydrolipoyl acetyltransferase protein (E2p) [39]. Our group has previously reported that a genetic fusion of a portion of the unstructured N-terminal region and D1 of Pfs230 to E2p increases the magnitude and longevity of the TRA [40].

*Helicobacter pylori* ferritin, referred to as ferritin hereafter, is a non-heme binding ferritin that controls iron storage within the bacterium and shares structural similarities with other bacterioferritins [41,42,43,44,45,46]. Ferritin forms a 24-subunit nanoparticle consisting of a 5-helix subunit where the first 4 helices form a bundle, and the 5th shorter helix is internal to the nanoparticle, forming stabilizing interactions within the nanoparticle [41,42,43,44,45]. Ferritin has 2-, 3-, and 4-fold symmetry resulting in 24-unit nanoparticles [41,42,43,44,45]. This symmetry is particularly useful for attaching antigens that require dimeric or trimeric interactions for the correct presentation of the antigen.

Ferritin has been used as a base for vaccines for a wide variety of disease-causing organisms including, but not limited to, viral [47,48,49,50,51,52,53,54,55,56,57,58,59,60,61,62,63,64,65,66,67,68,69,70] and bacterial [71,72,73,74] pathogens. Ferritin-based vaccines for human immunodeficiency virus-1 (NCT05903339), influenza A (NCT03186781, NCT03814720, NCT04579250, and NCT05155319), Epstein–Barr virus (NCT04645147 and NCT05683834), and SARS-CoV-2 (NCT04784767 and NCT06147063) are currently in clinical trials. Phase 1 clinical trials of multiple influenza A-ferritin-based vaccines (NCT03186781 and NCT03814720) have shown that the vaccines are safe and well tolerated. The proven safety of ferritin as a vaccine platform makes it a desirable system for use in the development of effective vaccines.

Here, we show that a Pfs230D1-ferritin fusion can be transiently expressed in a serum-free mammalian culture expression system. The Pfs230D1-ferritin nanoparticle forms stable and homogenous 24-copy nanoparticles. We also show that Pfs230D1-ferritin has a durable antibody response with high TRA when formulated with two different clinically relevant adjuvants. The results shown here form the basis for a readily manufactured single-component nanoparticle TBV, which has potent transmission-blocking activity.

## 2. Materials and Methods

### 2.1. Generation of Expression Plasmids

All constructs were synthesized and subcloned into pHLSec by Genscript. The Pfs230D1 construct was previously described and includes a portion of the N-terminal disordered region, amino acids 542–588, and D1, amino acids 589–736 (3D7 accession number XP_001349600.1) [40]. The sequence of our Pfs230D1 construct was also subcloned into a modified pHLSec plasmid containing the sequence for amino acids 3–167 of *Helicobacter pylori* (accession number WP_000949190.1) with two mutations, I7E and N19Q, and a linker with the sequence GGGGSGESQVRQQF that contains residues from the N-terminal extension found in bullfrog lower subunit ferritin [56]. The N19Q mutation was introduced into our construct to abolish a potential N-glycosylation site when ferritin is expressed in a mammalian expression system [56]. The I7E mutation was induced to establish a salt bridge that is conserved in bullfrog ferritin and interacts with the N-terminal extension [56]. This plasmid allows for the expression of a fusion protein consisting of Pfs230D1-linker-ferritin. The ferritin plasmid contains the ferritin sequence and linker described above with the addition of GGSGGS in the place of the D1 sequence.

### 2.2. Expression and Purification of Pfs230D1, Pfs230D1-Ferritin, and Ferritin

Using the commercially available Expi293F cell line obtained from Life Technologies ThermoFisher Scientific as part of the Expi293 expression system (Life Technologies ThermoFisher Scientific, Carlsbad, CA, USA), Pfs230D1, Pfs230D1-ferritin, and ferritin were transiently expressed for four days. The culture supernatant was harvested by centrifugation on day four post-transfection.

Pfs230D1 was then purified using Ni Sepharose excel resin (Cytiva, Washington, DC, USA) as previously described [40]. The Ni Sepharose excel resin purified protein was further purified using a Superdex 75 Increase 10/300 GL column (Cytiva) and phosphate-buffered saline (PBS).

The Pfs230D1-ferritin and ferritin were purified first by Q Fast Flow resin (Cytiva). The protein eluted from the Q Fast Flow resin was concentrated using a 50 kDa AMICON spin filter (EMD Millipore, St. Louis, MO, USA) as Pfs230D1-ferritin and ferritin self-assemble into higher-order 24-copy multimers that are well above the 50 kDa cutoff. The affinity-purified protein was further purified using a Superose 6 Increase 10/300 GL column (Cytiva). For protein used in functional assays and immunizations, the buffer used during size-exclusion chromatography was PBS. For protein used in negative-stain electron microscopy and stability assays, the buffer used during size-exclusion chromatography was 10 mM HEPES, pH 7.4, and 100 mM NaCl.

### 2.3. Negative-Stain Electron Microscopy and 2D Classification of Pfs230D1-Ferritin and Ferritin

Purified Pfs230D1-ferritin nanoparticles and ferritin nanoparticles at a concentration of 0.01 mg/mL in 10 mM HEPES, pH 7.4, 100 mM NaCl were individually adsorbed for 30 s on a glow discharged 300 mesh carbon-coated copper grids (Electron Microscopy Sciences, Hatfield, PA, USA) followed by methylamine tungstate (NanoW^®^—Nanoprobes Inc., Yaphank, NY, USA) staining. Raw micrographs were recorded using a Thermo Scientific Tecnai T20 microscope equipped with a charge-coupled device (CCD) camera. Using RELION 3.0, particles were auto-picked and averaged [75].

### 2.4. Stability Analysis of Pfs230D1-Ferritin

Six biological replicates of purified Pfs230D1-ferritin were split evenly between a time 0 and three treatment conditions: 24 h at 4 °C, 24 h at room temperature, and one round of freeze/thaw where the protein was stored at −80 °C. All samples were purified using a Superose 6 Increase 10/300 GL column (Cytiva) with 10 mM HEPES, pH 7.4, and 100 mM NaCl, and the amount of protein in the peak was determined using the Evaluation suite of UNICORN 7.3 software (Cytiva). The concentration of the protein in a given peak was determined by the Evaluation software 7.3 using the formula Concentration [mg/mL] = A/(d × 1000 × extinction coefficient), where A = average peak absorbance (Area/volume [mAu]) and d = UV cell path length [cm]. The percent recovered protein when compared to the time 0 sample was analyzed using GraphPad Prism 10 for MacOS.

### 2.5. Size-Exclusion Chromatography Small Angle X-ray Scattering (SEC-SAXS)

The Structurally Integrated Biology for Life Sciences (SIBYLS) beamline at beamline 12.3.1 was used for all SEC-SAXS experiments [76,77,78,79]. In brief, samples were injected onto a size-exclusion chromatography Shodex KW-804 column (Resonac Corporation, Tokyo, Japan) for Pfs230D1-ferritin or a size-exclusion chromatography Shodex KW-802.5 column (Resonac Corporation) for Pfs230D1, and the data for SAXS were collected as the protein came off the column with a λ = 1.03 Å incident light at a sample to detector distance of 1.5 m. Over the course of 40 min, a series of 3 s exposures were collected for each frame. SEC-SAXS data were analyzed using the ATSAS 3.2.1-1 software suite for macOS [80].

### 2.6. Ethics Statement

All rabbit studies were reviewed and approved under protocol LMIV 1E by the Institutional Animal Care and Use Committee (IACUC) at the National Institutes of Health (approval code: LMIV 1E and approval date: 1 November 2021) and performed in an American Association for Accreditation of Laboratory Animal Care (AAALAC)-accredited facility (AAALAC file #000777, last accredited in 2021). The PHS Animal Welfare Assurance (File Number # D16-00602) was last approved 30 May 2023.

Female New Zealand White rabbits approximately 11 weeks of age and weighing between 2.2 and 2.6 kg were obtained from Charles River Laboratories, pair-housed in 6-cage racks from Allentown, Inc. (Allentown, NJ, USA). Health monitoring was performed twice daily and rabbits were fed Rabbit Chow LabDiet^®^ 5321 (Richmond, IN, USA), given water ad libitum, with enrichment given 2–3 times per week. Studies were designed to reduce animal numbers where possible, and no more than momentary pain or distress was anticipated. All housing, husbandry practices, and pain management were in accordance with AAALAC guidelines, standards, and regulations.

### 2.7. Immunization of Rabbits with Pfs230D1, Pfs230D1-Ferritin, and Ferritin

For all Alhydrogel (Invivogen, San Diego, CA) formulations, a dose of 1 μg of each antigen was added to 1 mg/mL Alhydrogel, which contains a final dose of 500 μg aluminum hydroxide per injection, in Dulbecco’s Phosphate-Buffered Saline (DPBS), and incubated rotating for one hour at room temperature. For all AddaS03 (Invivogen) formulations, AddaS03 was added in a 1:1 volume to 1 μg of antigen for a final volume of 500 μL and mixed 10 times by pipetting. Between the time of formulation and injection into rabbits, all formulations were kept at room temperature. Rabbits were immunized subcutaneously in one site in the dorsal area with formulated vaccine as described above on day 0 and day 21. Rabbits immunized with Pfs230D1/AddaS03, ferritin/Alhydrogel, and ferritin/AddaS03 were bled for sera on days 0, 14, and 35 of the study. All other study groups were bled for sera on days 0, 14, 35, 63, 92, 119, and 147.

### 2.8. Determination of Pfs230D1 and Ferritin Specific Titers in Rabbit Sera Samples

Pfs230D1 or ferritin titers were determined by ELISA using a standard curve. Either Pfs230D1 or ferritin at a concentration of 0.02 mg/mL in Carbonate buffer, pH 9.6 was added to Nunc MaxiSorp ELISA plates (ThermoFisher) and incubated overnight at 4 °C. The following day, the plates were blocked for one hour at room temperature with 2% BSA in PBST. Diluted poly sera in 2% BSA and PBST were then added to the plates and incubated for one hour at room temperature. A 1:5000 dilution of anti-rabbit HRP conjugated secondary antibody (Bethyl Laboratories, Montgomery, TX, USA) was added to the plates and incubated for one hour at room temperature. The plates were washed three times with PBST between each of these steps and after the final incubation with the secondary antibody. The plates were developed with 3,3′,5,5′-Tetramethylbenzidine (TMB) Liquid Substrate System for ELISA (Millipore Sigma Aldrich, St. Louis, MO, USA) for 5 min for Pfs230D1 and 20 min for ferritin and the reaction stopped by the addition of equal volume of 2 M sulfuric acid for a final concentration of 1 M sulfuric acid in the wells. The plates were read with a Synergy H1 plate reader and 5gen v3.08 software, and the data were graphed and analyzed using GraphPad Prism 10 for MacOS.

### 2.9. Standard Membrane Feeding Assay

The SMFA was conducted as described previously [81,82]. In brief, total IgG from either day 35 or 147 was purified from individual rabbit poly sera and pooled. For the SMFA of day 35 samples, pooled IgG at concentrations of 3000 μg/mL, 1000 μg/mL, 333 μg/mL, or 111 μg/mL were mixed with ~0.2% of cultured stage V gametocytes (*P. falciparum* NF54 strain) in the presence of human complement (i.e., using non-heat-inactivated human serum) for a total volume of 260 μL (60 μL IgG, 100 μL human serum, 100 μL packed RBCs), and then fed to 3–6-day old female *Anopheles stephensi*. Deidentified Human O+ type sera and red blood cells for malaria cultures were obtained from GRIFOLS BioSupplies (Memphis, TN, USA). Eight days post-feeding, the mosquitoes were dissected to count the number of oocysts per midgut in 20 blood-fed mosquitoes. A second biological replicate for day 35 samples was conducted as described above using pooled IgG at concentrations of 333 μg/mL and 111 μg/mL. For the SMFA of day 147 samples, the SMFA was conducted as described above using pooled IgG at concentrations of 3000 μg/mL and 1000 μg/mL.

### 2.10. Statistical Analysis

Antibody Titers: The *p*-values for the Pfs230D1 specific titers for Pfs230D1/Alhydrogel vs. Pfs230D1-ferritin/Alhydrogel and Pfs230D1/AddaS03 vs. Pfs230D1-ferritin/AddaS03 for days 14 and 35 were determined using a Mann–Whitney test. The *p*-value for the Pfs230D1 specific titers at day 147 for Pfs230D1/Alhydrogel vs. Pfs230D1-ferritin/Alhydrogel vs. Pfs230D1/AddaS03 were determined by Kruskal–Wallis test. The *p*-values for the ferritin-specific titers for ferritin/Alhydrogel vs. Pfs230D1/Alhydrogel vs. Pfs230D1-ferritin/Alhydrogel and ferritin/AddaS03 vs. Pfs230D1/AddaS03 vs. Pfs230D1-ferritin/AddaS03 for days 14 and 35 were determined using a Kruskal–Wallis analysis followed by Dunn’s test to correct for multiple comparisons.

Stability Tests: The significance *p*-values for the paired data, Time 0 vs. 24 h at 4 °C, 24 h at room temperature, and a single cycle of freeze/thaw were determined using a Friedman test with a Dunn’s multiple comparisons test.

SMFA: The best estimate of TRA and the 95% confidence interval were calculated for each sample at each test concentration from two feeding experiments using a zero-inflated negative binomial (ZINB) model [82]. A significant difference between different IgG samples (at each concentration) was assessed using the same ZINB model, and Bonferroni correction was performed for multiple comparisons. The data were graphed and analyzed using GraphPad Prism 10 for MacOS or R version 4.2.1 (The R Foundation for Statistical Computing).

## 3. Results

### 3.1. Pfs230D1, Pfs230D1-Ferritin, and Ferritin Can Be Expressed and Purified from Mammalian Cells

Previously, we showed that a Pfs230D1 construct consisting of a portion of the N-terminal disordered region, amino acids 542–588, and D1, amino acids 589–736 could be expressed in Expi293F cells, a suspension mammalian cell line adapted from HEK293 cells, to a high level of purity [40]. In this study, we transiently transfected the same Pfs230D1 construct in Expi293F cells and harvested culture supernatants four days post-transfection. Pfs230D1 was purified from the supernatant by Ni^2+^ resin followed by size-exclusion chromatography (Figure 1B). The Pfs230D1 monomer eluted at 12 mL from a Superdex 75 10/300 increase size-exclusion column (Cytiva) and a single band of approximately 25 kDa on a reducing SDS-PAGE gel. This is consistent with the theoretical molecular weight of 21.8 kDa. Further analysis by Size-Exclusion Chromatography Small-Angle X-ray Scattering (SEC-SAXS) determined the molecular weight of 28.1 kDa (Table 1). The elution volume and SEC-SAXS calculated molecular weight are consistent with our previously reported elution volumes and SEC-SAXS molecular weight of 27.5 kDa [40]. The increase in apparent molecular weight of Pfs230D1 as determined by SEC-SAXS is also consistent with the known phenomenon that the molecular weight of unstructured proteins is often 1.2–1.8 times larger than the theoretical molecular weight [83,84] and the Pfs230D1 construct contains an unstructured N-terminal region.

*H. pylori* ferritin has been used as a self-assembling 24-copy vaccine platform by multiple groups and can be expressed in bacterial, mammalian, and insect cell expression systems [39]. We further expressed two ferritin constructs in a mammalian expression system. We transiently transfected Expi293F cells with the Pfs230D1-ferritin and ferritin constructs using plasmids encoding a secretion signal sequence ahead of the nanoparticle subunit. The supernatant was harvested four days post-transfection, and the nanoparticles were purified from the supernatant using Q Fast Flow ion exchange resin followed by size-exclusion chromatography (Figure 1C,D). The molecular weight of eluted Pfs230D1-ferritin in its multimeric form on the Superose 6 Increase 10/300 GL column (Cytiva) at approximately 12 mL is consistent with Pfs230D1-ferritin forming a higher-order multimer and had a single band on a reducing SDS-PAGE gel around 50 kDa (Figure 1C). The band observed on the SDS-PAGE gel is consistent with the theoretical molecular weight of the monomer of 42.6 kDa. The elution on the Superose 6 Increase 10/300 GL column is consistent with Pfs230D1-ferritin forming a higher-order multimer. To further examine the size of Pfs230D1-ferritin, we conducted SEC-SAXS on the purified sample. Pfs230D1-ferritin should theoretically form a 24-copy nanoparticle with a theoretical molecular weight of 1021.7 kDa. The molecular weight, as determined by SEC-SAXS, was 1065.3 kDa, consistent with the formation of a 24-copy nanoparticle (Table 1). The purified nanoparticle has yields averaging 70 mg/L culture.

The ferritin construct eluted at approximately 15 mL on a Superose 6 Increase 10/300 GL column, which is consistent with the formation of a higher-order multimer with a smaller molecular weight than Pfs230D1-ferritin (Figure 1D). A single band at approximately 20 kDa was observed on a reducing SDS–PAGE gel for ferritin, which was consistent with the theoretical molecular weight of 21.2 kDa for the ferritin monomer (Figure 1D). Further analysis of ferritin by SEC-SAXS determined the molecular weight in solution to be 521.7 kDa, which is consistent with the theoretical molecular weight of 508.3 kDa for a 24-copy particle (Table 1).

### 3.2. Pfs230D1-Ferritin and Ferritin form Uniform 24-Copy Nanoparticles

To examine sample quality, we conducted negative-stain electron microscopy of Pfs230D1-ferritin. Highly homogenous particles could be observed in the negative-stain images, and the 2D class averages clearly showed the uniformity of the particles (Figure 2A). The negative-stain images and 2D class averages are consistent with those of other *H. pylori* ferritin vaccines [51,66]. Similarly, the ferritin nanoparticles were found to be uniform particles by negative-stain electron microscopy (Appendix A). In summary, the SEC elution volume, SEC-SAXS calculated molecular weight, and the size of the particles observed by negative-stain electron microscopy were consistent with those of Pfs230D1-ferritin and ferritin self-assembling into 24-copy nanoparticles.

### 3.3. Pfs230D1-Ferritin Is Stable in Solution under Diverse Conditions

Vaccines need to be stable over time and under various conditions for successful mass administration. We evaluated the stability of our Pfs230D1-ferritin nanoparticles by exposing them to 24 h at 4 °C, 24 h at room temperature, and a single cycle of freeze/thaw after storage at −80 °C for 24 h. The samples were split evenly between a time zero control and the three test conditions for each independent replicate. The particle integrity of the time zero sample was analyzed by size exclusion immediately after splitting the biological replicate into the individual test conditions. The test condition samples were analyzed by size-exclusion chromatography after the test conditions were completed. Pfs230D1-ferritin was stable under all test conditions, with a percent protein recovery ranging from 81 to 104% after 24 h at 4 °C, 82 to 101% after 24 h at room temperature, and 93 to 100% after a single cycle of freeze/thaw (Figure 2B and Appendix A). While there was a significant difference between time 0 and free/thaw by a Friedman test with six repeat experiments (Appendix A), the percent recovery was always >93% in all experiments, suggesting the magnitude of the effect is minor.

### 3.4. A Low-Dose Two-Vaccination Regimen of Pfs230D1-Ferritin Is Sufficient to Produce High Antibody Titers against Pfs230D1

Female New Zealand White rabbits were immunized with 1 µg of either Pfs230D1, Pfs230D1-ferritin, or ferritin adjuvanted in either Alhydrogel or AddaS03 on days 0 and 21 of the study (Figure 3A). Alhydrogel, an aluminum hydroxide-based adjuvant, and AddaS03, an oil-in-water emulsion adjuvant and the research grade equivalent of AS03, were chosen because they are both clinically relevant adjuvants that induce distinct immune responses [85,86,87,88,89]. An additional consideration that was taken into account during the choosing of the adjuvants was that Alhydrogel has been tested in clinical studies with the Pfs230D1-EPA conjugate (clinical trial NCT02334462), which resulted in transmission-blocking antibodies [14,33].

Pfs230D1-specific antibody responses were evaluated two weeks post-vaccination on days 14 and 35 by ELISA (Figure 3B,C). Pfs230D1 and Pfs230D1-ferritin adjuvanted with Alhydrogel had no significant difference in Pfs230D1-specific titers at days 14 and 35 (*p* = 0.3939 at both timepoints) (Figure 3B). The median titers for day 14 were 3.3-fold greater for Pfs230D1-ferritin than for Pfs230D1, 2789 and 852, respectively (Appendix A). On day 35, the median titers for Pfs230D1 were 1.9-fold greater than that for Pfs230D1-ferritin, 67,107, and 35,080, respectively (Appendix A). The Pfs230D1-specific titers for Pfs230D1 and Pfs230D1-ferritin adjuvanted in Alhydrogel revealed that there was no significant difference in overall Pfs230D1-specific antibodies between the two groups. The Pfs230D1-specific antibody titers were also determined for ferritin adjuvanted with Alhydrogel, and as expected, there were no significant Pfs230D1-specific antibodies on either day 14 or 35 with titers at background levels (Figure 3B and Appendix A). Ferritin-specific antibody titers were also determined for all groups (Appendix A). The ferritin and Pfs230D1-ferritin groups adjuvanted with Alhydrogel-induced ferritin-specific antibodies, while the Pfs230D1 group adjuvanted with Alhydrogel had titer values at or below background levels (Appendix A).

In contrast, Pfs230D1 and Pfs230D1-ferritin adjuvanted with AddaS03 had markedly different Pfs230D1-specific antibody titers (Figure 3C). On both days 14 and 35, Pfs230D1-ferritin had significantly greater Pfs230D1-specific antibody titers than Pfs230D1 (*p* = 0.0022 at both timepoints) (Figure 3C). The median titers for Pfs230D1-ferritin were 147.9- and 2056.3-fold greater than those for Pfs230D1 on days 14 and 35, respectively (Appendix A). The Pfs230D1-specific median titers for Pfs230D1-ferritin adjuvanted with AddaS03 were 2958 and 228,250 for days 14 and 35, respectively (Appendix A). This significant difference in Pfs230D1-specific titers between Pfs230D1 and Pfs230D1-ferritin adjuvanted with AddaS03 may be attributed to the fact that AddaS03 is known to be a poor adjuvant for small proteins such as Pfs230D1. As expected, ferritin adjuvanted with AddaS03 had no significant Pfs230D1-specific antibodies on day 14 or 35, with titers being at or below background levels (Figure 3C). Ferritin-specific antibodies were also determined for all groups adjuvanted with AddaS03 (Appendix A). The ferritin and Pfs230D1-ferritin groups induced ferritin-specific antibodies, while the Pfs230D1 group did not with titers being at or below background levels (Appendix A).

### 3.5. Vaccination with a Low Dose of Pfs230D1-Ferritin Elicits a Durable Antibody Response

We evaluated the longevity of the Pfs230D1-specific antibody response. Serum samples for Pfs230D1 and Pfs230D1-ferritin adjuvanted in Alhydrogel and Pfs230D1-ferritin adjuvanted in AddaS03 groups were taken every 4 weeks for 112 days post day 35, with the final time point being day 147 (Figure 3D). The ferritin/Alhydrogel, ferritin/AddaS03, and Pfs230D1/AddaS03 groups were not followed past day 35 as they had minimal Pfs230D1-specific antibody responses on day 35 (Figure 3B,C and Appendix A). While Pfs230D1-specific titers decreased over time, all three groups retained high titers up to day 147 (Figure 3D,E). The median titers for Pfs230D1/Alhydrogel, Pfs230D1-ferritin/Alhydrogel, and Pfs230D1-ferritin/AddaS03 decreased 2.1-, 2.3-, and 6.5-fold, respectively, from day 35 to day 147 (Appendix A). On day 147, there was no significant difference in the Pfs230D1-specific titers among all three groups when analyzed by a Kruskal–Wallis test (*p* = 0.3263) (Figure 3E).

### 3.6. Low-Dose Pfs230D1-Ferritin Nanoparticle Vaccines Produced Potent and Durable Transmission-Blocking Activity

The standard membrane feeding assay is the gold standard for evaluating transmission-reducing activity [81]. Previous studies have shown that anti-Pfs230D1 mAbs require complement for activity [90,91,92] and all SMFAs in this study were conducted with human complement present. The total IgG was purified from pooled samples for each group and time point to reduce any potential background. Initially, one SMFA titration with purified IgG from Pfs230D1 and Pfs230D1-ferritin adjuvanted with either Alhydrogel or AddaS03 was conducted with a starting concentration of 3000 μg/mL and a 3-fold dilution series (Figure 4A and Appendix A). The IgG from Pfs230D1 adjuvanted with AddaS03 had the lowest TRA among the groups tested, which was consistent with the low Pfs230D1-specific titers determined for this group. Pfs230D1 and Pfs230D1-ferritin adjuvanted with Alhydrogel had similar TRA that were greater than those observed for Pfs230D1/AddaS03. However, the highest TRA activity was observed for Pfs230D1-ferritin adjuvanted with AddaS03. On day 35, IgG from the ferritin groups adjuvanted in either Alhydrogel or AddaS03 had <3% TRA at 3000 μg/mL (Figure 4A and Appendix A).

A second biological replicate of the SMFA was conducted at IgG concentrations of 333 μg/mL and 111 μg/mL to further evaluate the differences in TRA, and the best estimate of TRA and the 95% confidence interval were calculated for each sample at each test concentration from the two feeding experiments (Figure 4B; original values in each assay are seen in Appendix A). IgG from Pfs230D1 and Pfs230D1-ferritin adjuvanted with Alhydrogel showed 64.3% and 79.3% TRA at 333 μg/mL, respectively, and <−10% TRA for both at 111 μg/mL, and there was no significant difference between the purified IgG from the Pfs230D1/Alhydrogel and Pfs230D1-ferritin/Alhydrogel groups (*p* = 0.085 and 0.768, respectively). The two SMFA for the IgG from the Pfs230D1-ferritin group adjuvanted with AddaS03 showed consistently high TRA. Pfs230D1-ferritin had 99.5% and 88.4% TRA at 333 μg/mL and 111 μg/mL, respectively (Figure 4B). The TRA for Pfs230D1-ferritin/AddaS03 was significantly greater than that for Pfs230D1/AddaS03, which had <−3% TRA at both concentrations (*p* <0.001 for both concentrations).

To determine the durability of the TRA, two independent SMFA assays were conducted using purified IgG from day 147 pooled sera for Pfs230D1 and Pfs230D1-ferritin adjuvanted in Alhydrogel and Pfs230D1-ferritin adjuvanted in AddaS03 at concentrations of 3000 μg/mL and 1000 μg/mL, respectively (Figure 4C and Appendix A). These concentrations were chosen due to the decrease in Pfs230D1-specific titers from day 35 to day 147 (Figure 3D). At 3000 μg/mL, Pfs230D1-ferritin/Alhydrogel had a TRA of 93.3%, which was significantly greater than that of Pfs230D1/Alhydrogel, which had a TRA of 73.2% (*p* = 0.004) (Figure 4C and Appendix A). At 3000 μg/mL, Pfs230D1-ferritin/AddaS03 had a TRA of 98.0%, which was significantly greater than that of Pfs230D1/Alhydrogel (*p* = 0.002) and Pfs230D1-ferritin/Alhydrogel (*p* = 0.024) (Figure 4C). At 1000 μg/mL, Pfs230D1-ferritin/Alhydrogel had a TRA of 65.1%, which was significantly greater than that of the Pfs230D1/Alhydrogel, which had negligible TRA (*p* = 0.002) (Figure 4C). At 1000 μg/mL, Pfs230D1-ferritin/AddaS03 had a TRA of 69.9%, which was significantly greater than that of Pfs230D1/Alhydrogel (*p* = 0.002) but not that of Pfs230D1-ferritin/Alhydrogel (*p *> 0.999) (Figure 4C).

## 4. Discussion

Vaccines have proven to be highly effective at preventing or mitigating the severity of disease caused by several different pathogens, allowing for improved outcomes. The difficulty of vaccine development arises from several factors including the complexity of the disease targeted, the inability to express certain antigens at high yields and purities, and the lack of immunogenicity of some antigens. Malaria has suffered from all three of these issues. Several hurdles for a malaria vaccine include the multiple stages of the complex life cycle of *Plasmodium* parasites and identifying which antigen to target given that *Plasmodium* parasites are eukaryotic with a plethora of potential antigens. *Plasmodium falciparum* expresses more than 5000 proteins split across the three life cycle stages, and more than half of these proteins are hypothetical proteins with no known function [93].

In this study, we focused on the sexual/mosquito stage of the lifecycle and Pfs230, the most advanced of the TBV candidates [6,7,8,9,10,11,12,13,14,15,16]. Pfs230 is a ~300 kDa protein that has proven difficult to express and purify due to its large size, making the full-length protein a less-than-ideal vaccine candidate. As previous studies have shown, transmission-reducing activity has been isolated to a segment of the unstructured N-terminal region and D1 [11,14,26,28,29,30,31,32]. This shorter segment, which we termed Pfs230D1 in this study, can be produced much more easily than the full-length protein. Pfs230D1 has been expressed in mammalian culture systems, yeast, insect cell culture using baculovirus, and the wheat germ cell-free system. We chose the Expi293F cell expression system for the production of Pfs230D1 as it allows for the formation of the native disulfide bonds found in D1. We were able to produce highly pure Pfs230D1 for use in immunization studies using this system (Figure 1B), which was consistent with prior results [40].

Pfs230D1 has poor immunogenicity in AddaS03 and limited immunogenicity in Alhydrogel, two clinically relevant adjuvants (Figure 3B,C). The poor immunogenicity in AddaS03 was expected as AddaS03 has been shown to be a poor adjuvant for small proteins. AddaS03 is the research-grade equivalent of AS03 and a squalene-based oil-in-water emulsion adjuvant. In contrast, Alhydrogel is a wet gel suspension of aluminum hydroxide. Nanoparticles have been shown to improve the immunogenicity of vaccines through the repetitive display of antigens. To increase immunogenicity and simplify production, we genetically fused our Pfs230D1 construct to *H. pylori* ferritin (Figure 1C). The Pfs230D1-ferritin fusion developed here can be purified using a simplified tag-less scheme and is homogenous and stable. Critically, two low 1 μg doses of the nanoparticle in both adjuvants were sufficient to produce high titers of Pfs230D1-specific antibodies with high TRA (Figure 3 and Figure 4). This is in stark contrast to the Pfs230D1 monomer construct, which was unable to produce high titers of Pfs230D1-specific antibodies in AddaS03. Most importantly, the high TRA observed for the nanoparticles was maintained on day 147 of the study, which was 112 days (~4 months) post the last vaccination. The nanoparticle in AddaS03 elicited significantly greater TRA on day 147 at a concentration of 3000 µg/mL compared to the monomer and nanoparticle in Alhydrogel (Figure 4).

The differences observed on day 147 may be due to distinct modulation of the immune system by AddaS03 and Alhydrogel. Alhydrogel is known to cause antibody responses independent of Toll-like receptor signaling and to activate macrophages and dendritic cells. AddaS03/AS03 elicits a more balanced antibody response than Alhydrogel and induces cytokine and chemokine responses at the site of injection and in the draining of lymph nodes, enhancing antigen uptake by monocytes [88]. Taken together with known modes of immune system modulations, the data shown in this manuscript imply that adjuvant choice may play a critical role in long-term TRA, and further future studies to evaluate formulations in additional animal systems are warranted.

Disease-causing microorganisms are constantly evolving to evade host immune systems, and this is a consideration in the development of any vaccine. Our group recently published the human epitope map for individuals immunized with the Pfs230D1-EPA conjugate formulated in Alhydrogel and AS01 [33]. The human antibody epitope map revealed that transmission-reducing epitopes are localized to one face of Pfs230D1 and that ten polymorphisms were found within these epitopes [33]. Only three polymorphisms were shown to have any effect on antibody binding (G605S, G605R, and D714N), and the effect was limited to mAb 230AS-18 [33].

We previously showed that the genetic fusion of Pfs230D1 to E2p, a self-assembling 60-copy nanoparticle, increased the overall TRA and longevity of TRA. In this study, we expanded upon the concept of a genetic fusion of Pfs230D1 to self-assembling nanoparticles by creating a Pfs230D1-ferritin nanoparticle. The data presented here also showed that the Pfs230D1-ferritin nanoparticle increased overall TRA and longevity of TRA in two clinically relevant adjuvants, similar to what has been observed with Pfs230D1-E2p [40]. Both Pfs230D1-ferritin and Pfs230D1-E2p appear to have similar activity and production yields and both nanoparticle designs could be moved forward for future development. Further studies on final products produced in a manner suitable for use in humans will enable a direct comparison of the transmission-reducing activity of the two nanoparticles, establish if there are any differences in manufacturability and stability, and may identify a single candidate that is superior. These studies will pave the way to establish a next-generation transmission-blocking vaccine based on a Pfs230D1 nanoparticle. This study is limited in that it was performed once in rabbits, although it is the second study of a genetic fusion of Pfs230D1 to a self-assembling nanoparticle. A second limitation of this study is that it was not developed to unequivocally identify a single adjuvant for future use, although the data from day 147 indicate that AddaS03 may be a better adjuvant than Alhydrogel for Pfs230D1-ferritin. Finally, this study does not examine any potential role for cellular immunity induced by the nanoparticle or role in transmission-reducing activity. Future studies are needed to fully determine the best adjuvant for Pfs230D1-ferritin and the mechanism of induction of long-lived TRA.

This study expands on how genetic fusions of Pfs230D1 to self-assembling nanoparticles can improve immunogenicity and TRA over that of the Pfs230D1 monomer. Pfs230D1-ferritin can be easily purified in a simple purification scheme that has the potential to simplify the production of a Pfs230D1 vaccine when compared to Pfs230D1 conjugate vaccines. Simplifying production also has the benefit of reducing the cost of a TBV and potentially increasing the availability of TBVs to low-income countries.

## Figures and Tables

**Figure 1 vaccines-12-00546-f001:**
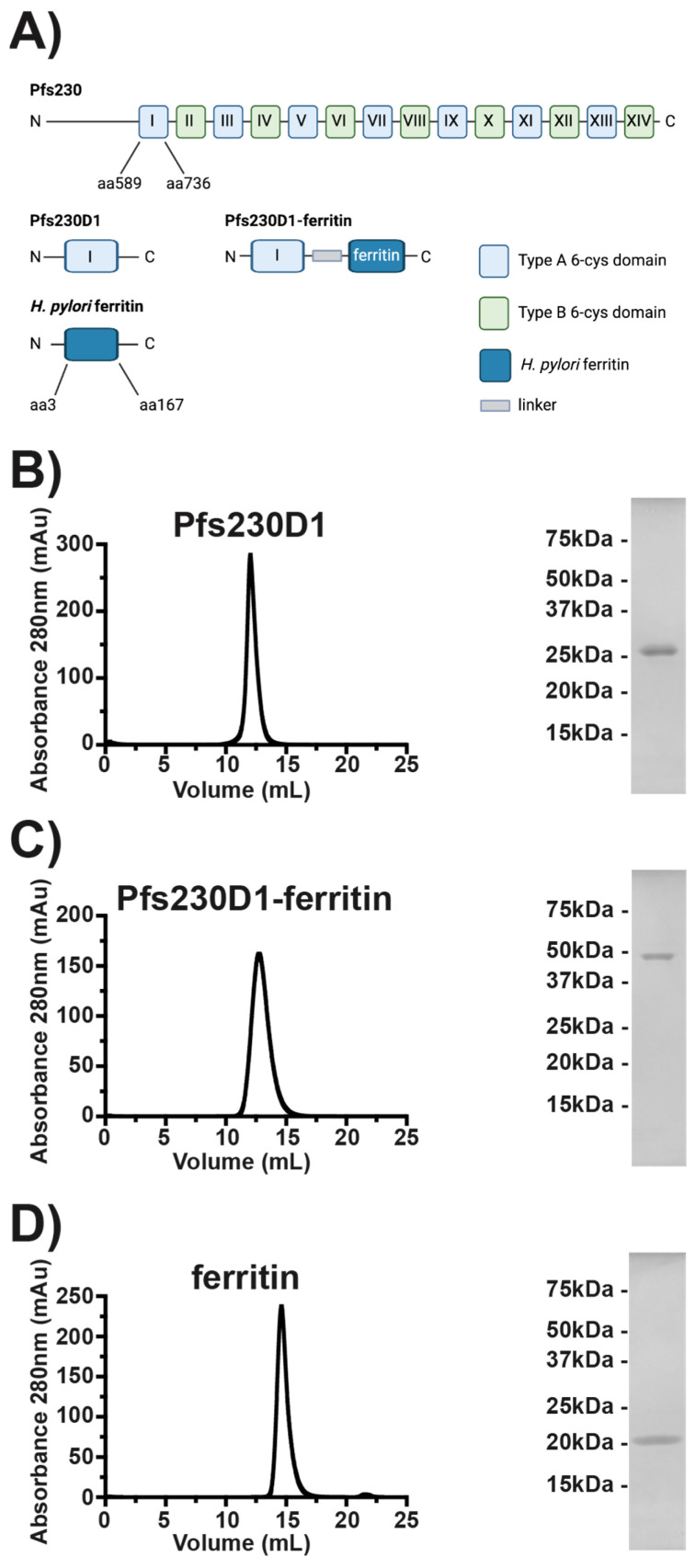
Pfs230D1-ferritin can be expressed in Expi293F cells: (**A**) Domain architecture of Pfs230, *H. pylori* ferritin, and Pfs230D1-ferritin constructs. Evaluation of the size and purity of (**B**) Pfs230D1 purified on a Superdex 75 10/300 increase size-exclusion column (Cytiva), (**C**) Pfs230D1-ferritin purified on a Superose 6 Increase 10/300 GL column (Cytiva), and (**D**) ferritin purified on a Superose 6 Increase 10/300 GL column (Cytiva) by size-exclusion chromatography and SDS-PAGE.

**Figure 2 vaccines-12-00546-f002:**
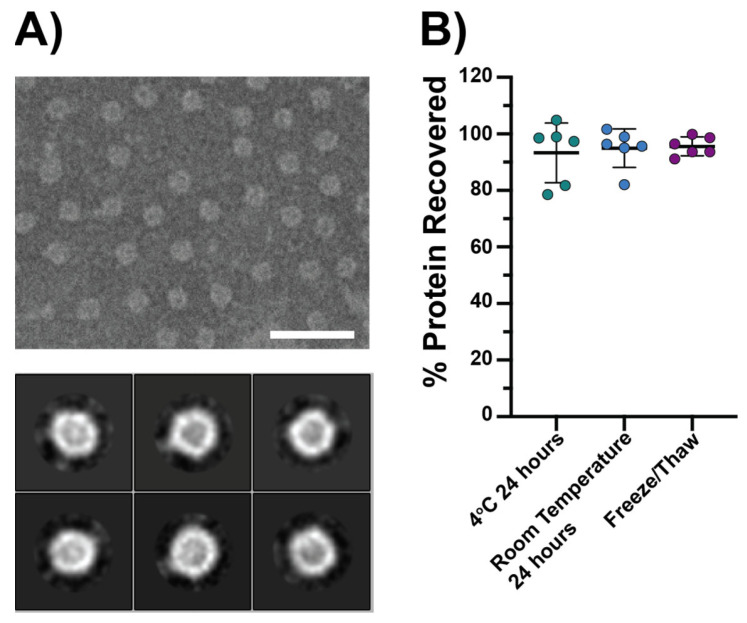
Pfs230D1-ferritin forms uniform nanoparticles that are stable: (**A**) Negative-stain electron microscopy image of Pfs230D1-ferritin (top) with 2D classification averages (bottom) scale bar = 50 nm. (**B**) Percent protein recovery of Pfs230D1-ferritin by size-exclusion chromatography after incubation at 4 °C or room temperature for 24 h or one cycle of freeze/thaw, median and 95% confidence interval shown from six independent replicates.

**Figure 3 vaccines-12-00546-f003:**
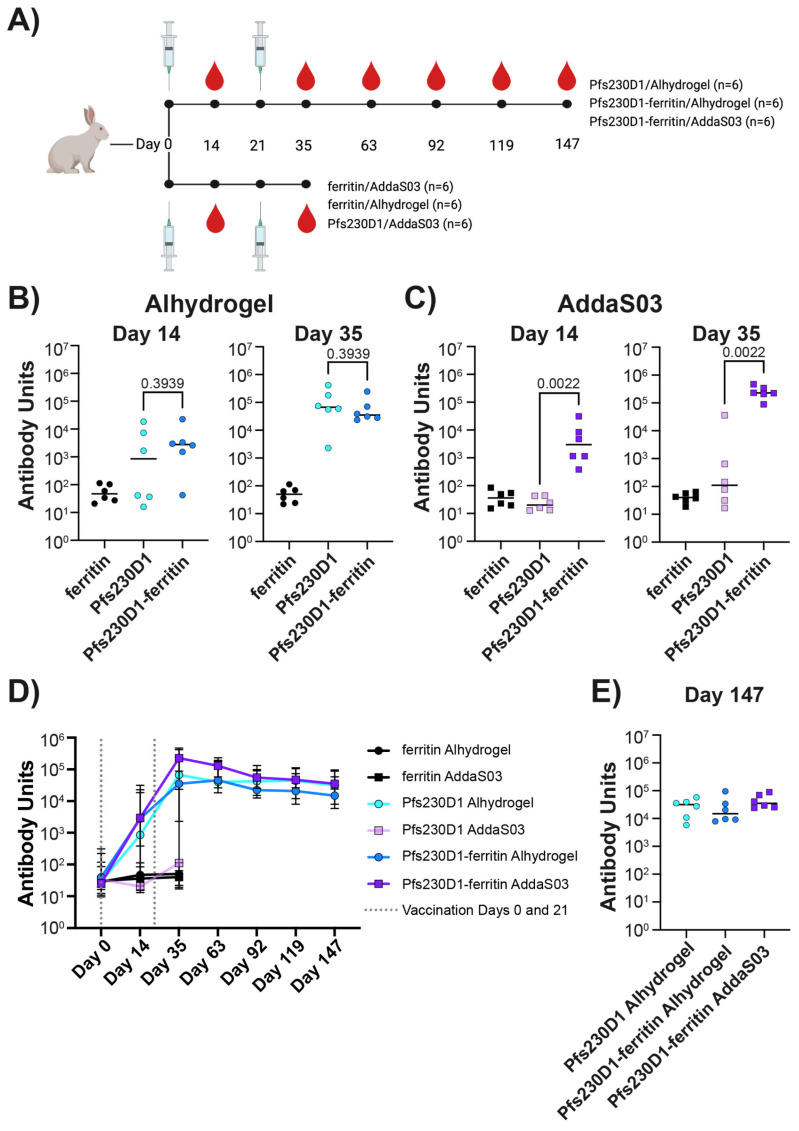
Pfs230D1-ferritin induces long-lived high levels of Pfs230D1 specific antibodies: (**A**) Timeline of immunizations. Pfs230D1 specific antibody titers for (**B**) day 14 and day 35 for groups adjuvanted with Alhydrogel, *p*-values were determined using a Mann–Whitney test, bars represent the median. (**C**) Day 14 and day 35 for groups adjuvanted with AddaS03, *p*-values were determined using a Mann–Whitney test, bars represent the median. (**D**) Pfs230D1-specific antibody titers for all groups over time, median, and 95% confidence interval shown. Vaccination days 0 and 21 are indicated by dashed lines. (**E**) Pfs230D1-specific antibody titers on Day 147 for Pfs230D1 and Pfs230D1-ferritin adjuvanted with Alhydrogel and Pfs230D1-ferritin adjuvanted with AddaS03; a Kruskal–Wallis test showed no statistical difference *p* = 0.3263, bars represent the median.

**Figure 4 vaccines-12-00546-f004:**
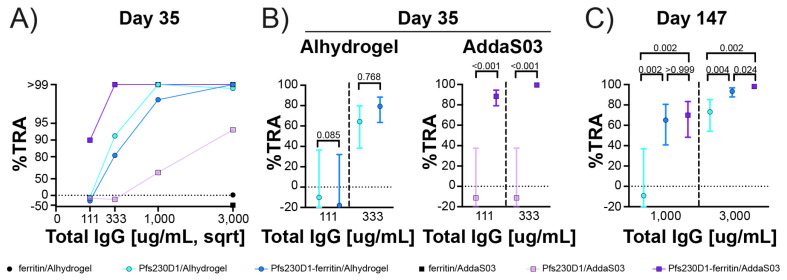
Fusion of Pfs230D1 to ferritin increases the potency and durability of transmission-reducing activity (TRA): (**A**) SMFA titration of purified IgG from day 35 rabbit sera immunized with Pfs230D1 and Pfs230D1-ferritin adjuvanted with Alhydrogel or AddaS03. (**B**) Best estimate of TRA and the 95% confidence interval (95%CI; error bar) from two independent SMFA for purified IgG from day 35 rabbit sera immunized with Pfs230D1 and Pfs230D1-ferritin adjuvanted with either Alhydrogel or AddaS03 are shown. The best estimate, 95%CI, and statistical significance were evaluated by a zero-inflated negative binomial (ZINB) model [82]. (**C**) Best estimate and 95%CI of TRA from two SMFA for purified IgG from day 147 rabbit sera immunized with Pfs230D1 and Pfs230D1-ferritin adjuvanted with either Alhydrogel or AddaS03 are shown. The best estimate, 95%CI, and statistical significance were evaluated by the same ZINB model, and Bonferroni corrected *p*-values are shown.

**Table 1 vaccines-12-00546-t001:** Theoretical and experimentally determined molecular weights of Pfs230D1, Pfs230D1-ferritin, and ferritin. Rg = radius of gyration.

	Subunit Theoretical Molecular Weight (kDa)	Theoretical Oligomeric State	Particle Theoretical Molecular Weight (kDa)	Molecular Weight Determined by SEC-SAXS (kDa)	Rg Determined by SEC-SAXS (Å)
Pfs230D1	21.8	monomer	-	28.1	24.62 ± 0.44
ferritin	21.2	24-copy	508.3	521.7	53.59 ± 0.61
Pfs230D1-ferritin	42.6	24-copy	1021.7	1065.3	82.48 ± 0.51

## Data Availability

The resources and materials used in the manuscript are available from the corresponding author upon request. Any such requests should be directed to and will be fulfilled by Niraj H. Tolia (niraj.tolia@nih.gov).

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
