# Peer review of "A Self-Assembling Pfs230D1-Ferritin Nanoparticle Vaccine Has Potent and Durable Malaria Transmission-Reducing Activity"

_vaccines, 2024, doi:10.3390/vaccines12050546_

Round 1
Reviewer 1 Report
Comments and Suggestions for Authors
Pleaser find below comments and suggestions for the manuscript entitled “A self-assembling Pfs230D1-ferritin nanoparticle vaccine has potent and durable malaria transmission reducing activity”.
Lines 21, 27:
Please use consistent terms (Line 21 - Single copy, Line 27 – single component). It appears line 21 also should be ‘single component’.
For clarity, it may also be useful to define for the first time as:
‘The single component Pfs230D1-ferritin construct forms stable, homogenous nanoparticles containing 24 copies of the Pfs230D1-ferritin fusion protein molecules’.
Materials and Methods:
Line 135: Pfs230D1, Pfs230D1-ferritin, ferritin, 230AL-26, 230AS-73, 230AS-18, and LMIV230-01 were transiently expressed…
Amend as: Pfs230D1, Pfs230D1-ferritin, ferritin, and human monoclonal antibodies 230AL-26, 230AS-73, 230AS-18, and LMIV230-01 were transiently expressed…
Line 144:
Since Pfs230D1-ferritin and ferritin proteins expressed using the Expi293 expression system (as per SDS-PAGE results) migrate at around ~48 kDa and ~21 kDa respectively in their reduced forms, it would be better to mention here that as Pfs230D1-ferritin and ferritin self assemble into 24-copy multimeric forms yielding higher molecular wirghts, a 50 kDa cutoff device was used to concentrate purified products.
Line 152: ….and bound IgGs were eluted off the column…
Line 157: Purified Pfs230D1-ferritin were individually adsorbed… (may need rephrasing)
Line 200: Define SIBYLS
Line 210: Alhydrogel/alum – Please use consistent adjuvant name.
Line 211: Define DPBS
Results:
Line 289: Table 1, Please define ‘Rg’
Line 294: We transiently transfected Pfs230D-ferritin and ferritin ….
Amend as: We transiently transfected Expi293F cells with Pfs230D-ferritin and ferritin constructs ….
Line 301:
The elution on the Superose 6 Increase 10/300 GL column is consistent with Pfs230D1-ferritin forming a higher order multimer.
May amend as: The MW of eluted Pfs230D1-ferritin in its multimeric form on the Superose 6 Increase 10/300 GL column is consistent with Pfs230D1-ferritin forming a higher order multimer.
Line 316:
The sub title 3.2 “Pfs230D1-ferritin forms a uniform 24-copy nanoparticle”.
Can be amended as “Pfs230D1-ferritin and ferritin form uniform 24-copy nanoparticles”.
Line 324-325:….Pfs230D1-ferritin self-assembling into 24-copy nanoparticles.
Amend as: …..Pfs230D1-ferritin and ferritin self-assembling into 24-copy nanoparticles.
Line 335: Section 3.3 Stability
What was the freezing temperature for freeze/thaw cycle? What is the intended storage temperature for bulk products? It may be mentioned if bulk purified product Pfs230D1-ferritin is being tested for extended storage at 4 C and/or frozen temperatures. Since the intended final formulations of Pfs230D1-ferretin are either with Alhydrogel or AddaS03, stability should be also be tested in formulated forms.
Line 353: Ref No 29 – Pls put as superscript.
Line 434: Please specify that IgGs were from Day 35 sera.
Discussion:
Since the antibody titers and TRA values of Pfs230D1-ferritin formulated with Alhydrogel and AddaS03 are very similar to that of Pfs230D1-E2p (also formulated with Alhydrogel and AddaS03) recently reported in Ref No 38, also briefly discuss any critical factors which may give advantage to a Pfs230D1-ferritin formulation for taking forward to advanced stages of testing.
Other:
Lines 96, 243, 246, 291, 321, 511:
Italicize (H. pylori, P. falciparum, Anopheles stephensi) as applicable in these lines.
Thank you.
Reviewer 2 Report
Comments and Suggestions for Authors
Salinas et al. generated a self-assembling Pfs230D-ferritin nanoparticle vaccine which can induced a decent humoral response to Pfs230 and the IgG collected from immunized rabbits offers significant transmission reducing activities. The authors used a similar concept that was published in NPJ vaccine last year (PMID: 37596283) to generate another version of Pfs230D vaccine. It has some novelty in design and shows the activities as predict.
Major concern:
For TRA studies, the authors used purified IgG from pooled sera to do the assays which limited the information about individual variation. Since the authors have the antisera, it will be informative that the authors can use 15ul or 30ul individual serum of D35 and D147 to do the TRA studies, which they have done in their NPJ vaccine paper.
For Figure 4B and 4C, it should be better to just present the individual result. It was a concern that the sample size is 2 or 3 with wide 95% confidence.
For Figure 4C, it will be more clear to label alhdrogel or AddsS03 on the top as Figure 4B.
For Figure 4, it will be important to state which statistic method was used to measure the difference in figure legend.
Reviewer 3 Report
Comments and Suggestions for Authors
The presented manuscript examines the vaccine potential of ferritin-antigen nanoparticles (Pfs230D1) for protection against malaria. The presented results are of interest to researchers in this field. The work was carried out at a high professional level and can be published in the Vaccines journal.
However, before publishing, I would like to clarify two questions:
1. Have the authors examined the formation of cellular immunity to Pfs230D1-ferritin in addition to the humoral response described in the manuscript?
2. From Figure 3B it can be seen that Pfs230D1-ferritin (with Alhydrogel adjuvant) induces faster antibody production than Pfs230D1. However, at day 35, there is a higher level of antibodies to Pfs230D1 than to Pfs230D1-ferritin. Does this mean that the use of Pfs230D1 is more preferable for the formation of long-term immunity?
Reviewer 4 Report
Comments and Suggestions for Authors
It is a nice work by a group with high experience on vaccination, based partly on their former results (Refs.29, 38). They described the development of a 24-copy self-assembling nanoparticle vaccine comprising domain 1 of Pfs230 genetically fused to H. pylori ferritin. They showed that the nanoparticle was highly immunogenic when rabbits were immunized. Both the methods and the results are clearly presented. There only minor problems with the presentation that can be easily corrected.
Minor problems:
Species and genus names should be italicized.
line 128: Question: is any significance of these mutations or are they mentioned only for the records?
line 135: It is disturbing that the reader does not know what 230AL-26, 230AS-73, 230AS-18, and LMIV230-01 are. They are defined much later, in lines 355-356. I suggest defining them either in line 132 or in line 135.
line 198: Graphpad instead of Grapahpad
lines 282-288. The apparent MW of the Pfs230D1 monomer is on SDS gel and by SAXS 25 kDA and 28.1 kDA, respectively. These values are somewhat higher than the theoretical MW (21.8) (15%, 28%). It can be discussed that the reason is that Pfs230D1 contains a segment of the unstructured N-terminal region of the protein. It is known that the apparent MW of intrinsically unstructured proteins is often 1.2–1.8 times higher than the real one [Pascal et al. Protein Expression and Purification 47 (2006) 524–532; Tompa, Trends Biochem. Sci. 27 (2002) 527–533.].
References:
Species and genus names should be italicized also in References.
Some references are incomplete or incorrect:
Ref 14:
Clin Invest 2021, 131 (7), e146221.
Ref 29:
Immunity 2023, 56 (2), 433-443 e5.
Ref 64:
Sci Immunol 2020, 5 (47), eaba6466.
Ref 65:
Viruses 2022, 14 (3), 541.
Ref 68:
Vaccines (Basel) 2023, 11 (4), 821
Ref 71:
Int J Mol Sci 2023, 24 (7), 6183.
Ref 73:
Elife 2018, 7, e42166.
Round 2
Reviewer 2 Report
Comments and Suggestions for Authors
All the concerns have been addressed.
Author Response
Thank you